# Real-Time 6-DOF Pose Estimation of Known Geometries in Point Cloud Data

**DOI:** 10.3390/s23063085

**Published:** 2023-03-13

**Authors:** Vedant Bhandari, Tyson Govan Phillips, Peter Ross McAree

**Affiliations:** School of Mechanical and Mining Engineering, The University of Queensland, Brisbane, QLD 4072, Australia

**Keywords:** pose estimation, real-time, perception, point cloud

## Abstract

The task of tracking the pose of an object with a known geometry from point cloud measurements arises in robot perception. It calls for a solution that is both accurate and robust, and can be computed at a rate that aligns with the needs of a control system that might make decisions based on it. The Iterative Closest Point (ICP) algorithm is widely used for this purpose, but it is susceptible to failure in practical scenarios. We present a robust and efficient solution for pose-from-point cloud estimation called the *Pose Lookup Method* (PLuM). PLuM is a probabilistic reward-based objective function that is resilient to measurement uncertainty and clutter. Efficiency is achieved through the use of lookup tables, which substitute complex geometric operations such as raycasting used in earlier solutions. Our results show millimetre accuracy and fast pose estimation in benchmark tests using triangulated geometry models, outperforming state-of-the-art ICP-based methods. These results are extended to field robotics applications, resulting in real-time haul truck pose estimation. By utilising point clouds from a LiDAR fixed to a rope shovel, the PLuM algorithm tracks a haul truck effectively throughout the excavation load cycle at a rate of 20 Hz, matching the sensor frame rate. PLuM is straightforward to implement and provides dependable and timely solutions in demanding environments.

## 1. Introduction

In field robotics, estimating the pose (position and orientation) of known geometries within point cloud data is a common challenge. The motivation is to provide robotic agents with the perceptual information to detect and interact with objects in their immediate environment. In 3D environments, this involves determining the (x,y,z)-translations and (roll,pitch,yaw)-rotations that position and orient the object relative to a known reference frame. Alternatively, this may amount to the estimation of kinematic parameters, whereby the geometry is constrained to articulate via kinematic linkages, e.g., six rotational joint angles of a 6R robot. The objective is to find the best-matching frame transform between the known geometry and the observed point cloud measurements.

There are two key requirements for pose-from-point cloud algorithms: (i) the solution must be sufficiently accurate and robust to meet the demands of the problem, and (ii) the solution needs to be computed at a rate suitable for control decisions. There is a wealth of literature on pose estimation solutions, with most tailored to the use case they refer to. We hold that a desirable pose estimation algorithm should be agnostic to the use case and require minimal configuration.

This paper presents a solution that is straightforward to implement, does not require configuration, and meets the challenges of accuracy, robustness, and timeliness for a broad class of scenarios. The method is based on similar concepts presented in our previous work [1,2]. The issues of reliability, accuracy, and robustness are addressed by utilising reward-based metrics. The challenge of real-time computation is resolved by replacing complex geometric operations, such as raycasting, with pre-calculated lookup tables. We refer to this method as the *Pose Lookup Method* (PLuM).

## 2. The Challenges of Estimating Object Pose in Point Cloud Data

Point clouds are collections of range measurements obtained at specific heading and elevation angles. These measurements lack semantic information and a relationship with the environment they represent beyond geometric considerations. Deriving meaningful information from raw point cloud data autonomously is a process that is fundamentally limited.

Pose-from-point cloud algorithms need to meet two fundamental requirements that present challenges: (i) the provision of reliable solutions and (ii) the provision of timely solutions.

### 2.1. Challenge 1: Providing Reliable Solutions

A reliable pose-from-point cloud algorithm is one that is consistently accurate and precise irrespective of the point cloud data that it is presented with. An unreliable solution is characterised by the divergence of the computed estimate from the true pose under different conditions.

Solutions diverge for many reasons. The Cartesian point measurements of the cloud, for example, may be in error or they may be affected by confounding factors. Issues may arise, for example, from airborne particles that inhibit the sensor’s ability to measure objects in the field of view [3]. There are chained errors that accumulate. Per-beam intrinsic parameters are only known to a degree of certainty that is often obtained through calibration procedures [4,5,6]. If the sensor is installed on a robotic platform, it is usually necessary to know its location relative to a navigation reference frame on that platform and small orientation errors translate to large point cloud errors at long range. Extrinsic registration procedures to determine this have inherent uncertainty [7,8]. The challenge of reliability is magnified in field robotics environments, which are cluttered, unordered, and can be in a constant state of unpredictable flux [9].

To be reliable, cost-based algorithms, such as Iterative Closest Point (ICP), require the segmentation of points in the cloud associated with the object of interest [10]. This usually requires a preprocessing step, often individually tailored to the application. It is for this reason that we previously pursued reward-based solutions that do not require segmenting of complex point cloud measurements [2].

### 2.2. Challenge 2: Providing Timely Solutions

Many field robotics applications require pose estimates within the time span of discretion of the process that uses them. Many existing algorithms require significant preprocessing, configuration tuning, or complex geometric calculations, and are unable to provide a real-time solution. For example, our previous work in [11] presented a robust and accurate pose estimation method with a high computational expense. We reported 99.5% of the total execution time being utilised by the raycasting operation in the objective function.

A bottleneck for pose-from-point cloud algorithms is searching the 6-DOF space for the optimal model pose that best matches the sensor data. Aiming for millimetre accuracy in the translations and 1° accuracy in the rotations in a 3m×3m×3m space results in $1.2597\times 10^{18}$ possible hypotheses. This problem is coupled with the increasing amount of data available for interpretation from modern sensors at high frequencies. For example, the Velodyne ULTRA Puck provides up to 600,000 points per second at frame rates between 5 and 20 Hz [12], and the Ouster OS-128 provides up to 2,621,440 points per second at frame rates between 10 and 20 Hz [13]. It is challenging to interpret and search the hypothesis space at speeds in accordance with the sensor frame rate. Recent advancements in hardware have allowed for faster computation, aiding in the development of faster algorithms.

However, the requirement of a fast objective function still needs to be satisfied. Parallel computing architectures such as OpenCL [14] and NVIDIA’s CUDA toolkit [15] allow for evaluating multiple hypotheses in parallel, decreasing the overall time to determine a pose estimate. These tools are taken advantage of for the real-time implementation of PLuM.

## 3. Related Work

The most common method for solving the model-based pose estimation problem, also known as geometric registration, is the Iterative Closest Point (ICP) algorithm introduced by Chen and Medioni [16] and Besl and McKay [10] in the early 1990s. ICP begins with an initial alignment estimate between the model and the point cloud. Assumedly correct correspondences are then iteratively assigned between the points of both datasets while determining a transformation that minimises the $l_{2}$ difference between them. ICP has six key steps: selection, matching, weighting, rejecting, error metric selection, and minimising, as detailed in [17]. The vanilla ICP algorithm handles 6-DOF pose estimation, is independent of shape representation, and can handle a reasonable amount of normally distributed vector noise [10], but with severe limitations. The pose estimation results depend strongly on the initial alignment, require point cloud segmentation, and tend to converge to local extrema. The algorithm is sensitive to real-world scenarios, such as noisy data and clutter, and considers the removal of statistical outliers as a preprocessing step.

The problem of locating a known geometry in point clouds arises in many applications. A common use is to perform digital quality inspection by registering physical components to a ground truth CAD model to highlight manufacturing defects, as demonstrated by [18]. Using a high-resolution point cloud sensor allows for quality inspection agnostic to the CAD model, with Coordinate-Measuring Machines (CMMs) unable to assess the quality of complex free-form surfaces such as boat propellers. Three-dimensional shape registration also has versatile applications in bioinformatics and the medical imaging industry, with the point cloud consisting of pixel values instead of ranges. For example, a common problem in drug design involves the use of 3D shape registration for determining the optimal transformation that superimposes active regions of two known protein structures to provide a comparison of protein binding sites. Ellingson et al. [19] provided an application example, proposing a new algorithm based on the Iterative Closest Point method for accurate protein surface matching, whereas Bertolazzi et al. [20] focused on a more timely solution by increasing the efficiency of the optimisation search. Another example is an ICP-inspired algorithm for retinal image registration [21]. Robotics has also observed an increasing application of registration problems prominent in pick and place robots such as in [22], demonstrating the ability to sort products with a random pose on a conveyor belt. The mining industry uses registration in an attempt to automate the stages of the excavator loading cycle. Borthwick demonstrated a method for haul truck estimation and load profiling using stereo vision and the ICP registration algorithm [23], whereas Cui et al. [24] presented a registration network for excavator bucket pose estimation. Phillips et al. demonstrated a method for both haul truck and dipper pose estimation using a LiDAR sensor and the Maximum Sum of Evidence method [11]. In the various applications mentioned, algorithms tend to focus on either the solution’s robustness or the solving efficiency. The gap for a method that provides both robustness and timeliness is apparent.

The research space is consumed with various enhancements of one or more of the six stages of the generic ICP algorithm mentioned above. Due to its simple formulation, the algorithm only provides ideal results in specific conditions, leading to the need for use-case-tailored variants. Pomerleau et al. [25] provided an overview of the legacy of ICP and showed the exponentially increasing number of papers published yearly since 1992, updated and displayed in Figure 1. Furthermore, Donoso et al. identified 20,736 variants of ICP achieved through combinations of the six steps of ICP [26]. Rusinkiewicz et al. [17] explored and demonstrated the derivation of variants from existing variants for better performance. The problem is apparent as variations are developed to handle edge cases as they arise. The general recommendation of searching for a variant that best suits the task and data, and if not, deriving a new variant, is not ideal. Regardless, the performance is not generally robust.

Common methods for solving the point cloud-to-model registration problem are summarised in Table 1 and discussed below.

The approach adopted in this paper is to provide a single solution that is not burdened by configuration. Variants are in active development using the ICP framework and improving certain aspects. Sparse ICP [27] received attention as it increased performance on datasets affected by noise, outliers, and incomplete data. However, it has a high computational expense and converges to local extrema when the two datasets are initialised too far from each other. Yang et al. [28] published Go-ICP, the first globally optimal algorithm providing superior results where a good initialisation is not available, but its usage is limited to scenarios where real-time performance is not critical. Zhang et al. [29] described a fast and robust implementation of ICP, but noted that good performance relies on good initialisation. This is a recurring theme in many variants where an improvement in one or more of the ICP steps compromises the performance of another, and this motivates the exploration of further variants.

Another direction that has been pursued to formulate point cloud registration is as a maximum likelihood estimation problem. Stemming from the vanilla ICP algorithm, a class of algorithms was introduced using statistical/probabilistic methods instead of pure error-based minimisation. The Coherent Point Drift (CPD) method by Myroneko and Song [30] considers the alignment of two point sets as a probability density estimation problem, with many similar publications modelling the registration task as a maximum likelihood estimation problem, including [31,32,33]. These methods overcome many limitations of the ICP algorithm, including the requirement of a good initial guess and occlusions in the scene, and provide superior pose estimation performance in the presence of noise, outliers, and missing points due to the method being reward-based and not minimising error. However, the algorithms tend to converge to local extrema, providing a non-optimal solution.

The Maximum Sum of Evidence method introduced by Phillips et al. [2] provides a robust probabilistic approach for model-to-point cloud registration, capable of determining the 6-DOF pose in noisy and occluded environments with no initial alignment. The method is robust, but computationally expensive due to the need to perform raycasting operations across multiple alternative hypotheses.

Advancements in other research fields continue to develop and improve the ICP algorithm. Genetic algorithms have been employed to determine global extrema and provide accurate results by extending the vanilla ICP algorithm, as in [37]. Another subdivision of the research space has spawned with the recent developments in deep learning and the introduction of geometric deep learning [34], with recent effective registration extensions in [35,36]. New sensing technologies such as Aeva’s Doppler 4D LiDAR [38] have also prompted improvements in ICP for point cloud-to-point cloud registration with the Doppler Iterative Closest Point [39], which leverages additional metadata in the returned point cloud such as per point instantaneous velocities for accurate registration. This has been a recurring theme noted by [25], with an increase in the application of the ICP algorithm fuelled by advancements in sensing and computing technology such as LiDAR, SLAM methods using range scanners, computer vision with less expensive and accessible cameras, and more recently, the surge of deep learning [40,41].

## 4. Problem Formulation

PLuM is an approximation of the Bayesian-based pose estimation algorithm called the Maximum Sum of Evidence (MSoE) method proposed by the authors in [2]. The reward metric of MSoE is based in the range space (i.e., comparing raw range measurement observations against raycast measurements under the assumed hypothesis). Here, we take advantage of a Euclidean-based reward metric allowing for real-time hypothesis evaluation. Model-based registration aims to find an optimal homogeneous transform, denoted $T_{S\to M}^{🟊}$, that maximises the likelihood of producing point cloud measurements observed in the sensor frame (S), as illustrated in Figure 2. This section uses a 2D model to introduce the algorithm and is easily extended to 3D for solving 6-DOF problems.

Figure 3 provides an overview of the entire process. The following provides a detailed explanation of the generation of the lookup table, the design of the objective function, and the selection of the pose hypothesis search algorithm.

PLuM uses a reward-based objective function to address the challenge of providing a reliable solution. Measurements are rewarded when they are likely to exist under an assumed sensor to model pose estimate. Using a reward-based metric instead of an error-based metric allows for robustness against sensor noise, clutter, outliers in the point cloud data, and any model mismatch in the known geometric model.

Given a set of *n* Cartesian point cloud measurements in the sensor frame, $P_{S}=\left\{p_{S,1},p_{S,2},\ldots ,p_{S,n}\right\}$, the reward, $r_{i|j}$, of observing the *i*-th sensor measurement given the *j*-th pose hypothesis, ${\hat{T}}_{S\to M,j}$, is calculated using a zero-mean Gaussian function of the minimum distance between the measurement and the known model, $d_{i|j}$, with the standard deviation given by the sensor’s measurement uncertainty, σ. This is visualised for a single sensor measurement in Figure 4 and mathematically described as,
(1)$$r_{i|j}=N\left(d_{i|j}|0,\sigma_{reward}\right)=\frac{1}{\sqrt{2\pi \sigma^{2}}}\exp\left(-\frac{d_{i|j}^{2}}{2\sigma^{2}}\right)$$

The reward function decreases as the distance to the model’s surface, $d_{i|j}$, increases. The standard deviation, σ, can be varied to change the rate at which the reward decreases. This is a configurable parameter of the objective function and is shown to be insensitive in the following Results Section. The normalisation constant, $1/\sqrt{2\pi \sigma^{2}}$, can be discarded as this scalar value will not affect the relative reward between hypotheses.

The total reward is calculated as the summation of all *n* points in the point cloud. The optimal hypothesis, $H^{🟊}$, is chosen as the hypothesis that maximises the sum of the reward.
(2)$$H^{🟊}=\max_{j}\left\{\sum_{i=0}^{n}r_{i|j}\right\}$$

The evaluation of a hypothesis is costly due to the calculation of point-to-model distances, $d_{i|j}$. Given *m* pose hypotheses to be fit to *n* point cloud measurements over *k* iterations of the objective function results in m·n·k costly point-to-surface calculations. The computation time is considerably decreased by using a lookup table of pre-calculated surface distances or corresponding rewards. Furthermore, the computation is not affected by the complexity of the geometry, e.g., the number of triangles typically extends the raycasting time, as reported by [42]. The distances, $d_{i|j}$, of all points in the vicinity of the model are precalculated. The rewards, $r_{i|j}$, are then calculated and stored via a lookup table implementation. Figure 5 depicts the lookup table in 2D.

Algorithm 1 describes the generation of the lookup table.
**Algorithm 1:** Lookup table generation (offline)
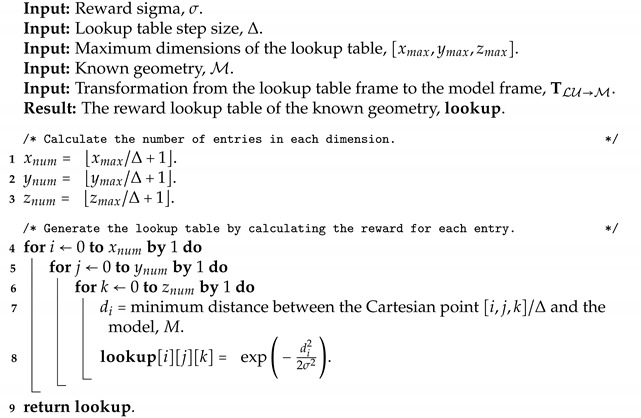


The *i*-th point in the sensor frame, $p_{S,i}$, is calculated in the lookup frame as,
(3)$$p_{\mathrm{LU},i}=T_{\mathrm{LU}\to S}p_{S,i}={\left[\begin{array}{c} x_{i}\\ y_{i}\\ z_{i}\\ 1 \end{array}\right]}_{\mathrm{LU}}$$

The frame transform between the lookup and sensor frame, $T_{\mathrm{LU}\to S}$, is calculated using the *j*-th hypothesised model pose, ${\hat{T}}_{S\to M,j}$, and the rigid transform to the lookup table, $T_{M\to \mathrm{LU}}$, as follows,
(4)$$T_{\mathrm{LU}\to S}={\left({\hat{T}}_{S\to M,j}T_{M\to \mathrm{LU}}\right)}^{-1}$$

With the sensor point now known in the lookup table frame, the reward value is looked up by scaling the Cartesian coordinate of the point $\left(x_{i,\mathrm{LU}},y_{i,\mathrm{LU}},z_{i,\mathrm{LU}}\right)$ with the discretisation resolution, Δ, that was used to construct the table.
(5)$$r_{i|j}=\mathrm{lookup}\left(\left\lfloor \frac{x_{i,\mathrm{LU}}}{\Delta }\right\rfloor ,\left\lfloor \frac{y_{i,\mathrm{LU}}}{\Delta }\right\rfloor ,\left\lfloor \frac{z_{i,\mathrm{LU}}}{\Delta }\right\rfloor \right)$$
where ⌊·⌋ represents the floor function, here used to obtain integer indices for the lookup table. The lookup table indices can also be found using the rounding function. However, truncation is faster than rounding, and a small resolution of Δ will result in a good approximation.

The reward calculation described is an objective function that ranks hypotheses. There are many search algorithms that can be used to find the maximum reward hypothesis. Gradient methods such as least squares or Gauss–Newton and simplex methods such as Nelder–Mead have been shown to be susceptible to local extrema. The search algorithm is independent of the objective function, as displayed in Figure 3. Our previous work in [2] provided a detailed investigation into the evaluation of different search algorithms. All results in the following section use the particle filter search algorithm, as it allows for fast and accurate determination of the global maxima in multiple dimensions. The algorithm’s search space and size are adjusted based on the problem.

The objective function is summarised below in Algorithm 2.
**Algorithm 2:** Pose Lookup Method (PLuM) algorithm—objective function
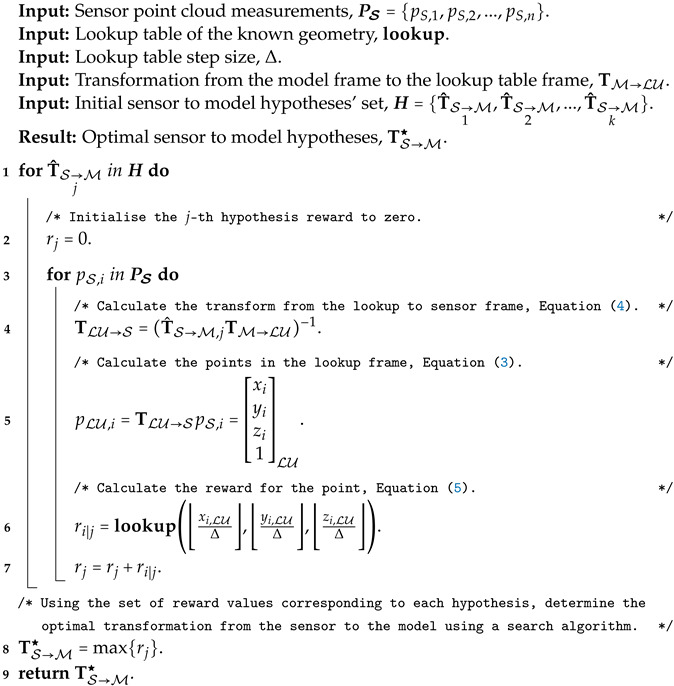


## 5. Results

The following section demonstrates the performance of the PLuM algorithm in various scenarios. PLuM is compared to other methods, including vanilla ICP [10], Fast and Robust ICP [29], and Sparse ICP [27], in overcoming the two critical challenges of pose estimation: providing a reliable and timely solution. The implementation of other methods is directly used from [43], which are CPU-threaded using OpenMP. The Stanford Bunny and Dragon models are used in this section to perform benchmark tests. A CPU and GPU-accelerated version of the PLuM algorithm is implemented for comparison. The GPU version is accelerated using CUDA in C++. The Maximum Sum of Evidence method previously demonstrated by the authors in [2] is not displayed due to its high computational expense, unable to satisfy the timeliness criterion. However, the results in Section 6 are compared against the original dataset used to demonstrate MSoE in [11]. All testing was conducted on an Intel i7 CPU @ 3.80 GHz with 62.5 GiB memory and an NVIDIA GeForce GTX 3090 GPU running Ubuntu 20.04.4 LTS.

The timeliness of the registration algorithms was recorded as the time taken for the execution of the algorithm from when it received the point cloud to the determination of the converged pose solution. The timing functions were used directly from the other implementations. The accuracy of a pose estimate is reported as the maximum vertex error, $e_{max}$, between the pose estimate and the ground truth, as illustrated in Figure 6. Six-degree-of-freedom problems are demonstrated below with the goal of estimating the six homogeneous parameters, including the rotations roll–pitch–yaw and the translations x-y-z represented as a row vector, ${\hat{T}}_{S\to M}\;=\;\mathrm{hom}\left([\theta ,\phi ,\psi ,x,y,z]\right)$.

### 5.1. Influence of the Initial Pose Estimate

Many pose estimation algorithms require some *good* initial guess for an accurate registration result. Figure 7 displays two candidate problems.

Other methods such as ICP provide acceptable performance in the presence of a perfect point cloud and some overlap between the initial guess and the true position of the model. Figure 7a displays a perfect point cloud of the Stanford Bunny model with 202 measurements, a true pose of $T_{S\to M}^{🟊}=\left[1.57,0,0,0.18,0.16,0\right]$, and an initial pose of ${\hat{T}}_{S\to M}=\left[1.17,-0.1,0.3,0.16,0.15,-0.01\right]$. Table 2 and Figure 8 report the accuracy and execution time results of the state-of-the-art methods in comparison to the proposed PLuM algorithm. The vanilla ICP algorithm reported a significant vertex error of 27.98 mm, whereas Robust ICP provided the smallest vertex error of 0.14 mm at the expense of being approximately 130 times slower than PLuM, which provided an error of 3.45 mm. PLuM demonstrated its ability to provide both accurate and timely results. Fast ICP failed due to insufficient overlap between the point cloud and initial guess, which is a recognised limitation by the authors [29]. Sparse ICP provided an answer with similar accuracy to PLuM, but with a significant computation time.

The same problem with an initial pose of ${\hat{T}}_{S\to M}=\left[1.05,0.5,1.57,0.06,0.17,-0.05\right]$ is displayed in Figure 7b. No overlap is present, with significant deviations in all six homogeneous transformations. This is a known problem for ICP-based methods and is observed in the results reported in Table 3 and Figure 9. All methods except PLuM were unable to perform the registration. PLuM provided a comparable vertex error of 3.89 mm to the previous result with some overlap and a timely result in 32 ms. Many applications cannot provide a *good* initial pose estimate, requiring significant algorithm configuration or the creation of a custom ICP variant. The proposed algorithm demonstrated robustness against the initial pose estimate.

### 5.2. Clutter Test

The results presented in the previous section used perfect point cloud measurements with no noise or spurious data. Real-world environments often contain other objects in the scene and spurious measurements. This can be effectively represented as clutter or randomly generated measurements in the area of interest. Methods such as ICP have severe limitations with clutter and often require segmentation before registration. We have thoroughly discussed the limitations of using error-based metrics in such situations in [2]. Many authors exploit probabilistic approaches to overcome the limitations of error-based metrics, such as [44], using a correlation-based registration approach, and [45], representing point sets as a mixture of Gaussians.

Figure 10 displays four scenarios of estimating the pose of the Stanford Dragon model with increasing clutter in the scene. The clutter-free point cloud has 187 measurements (green), whereas the scenario with 90% clutter has an additional 1683 random measurements (red). The clutter is randomly generated within a defined area around the dragon.

The maximum vertex error for up to 90% clutter using the PLuM algorithm was less than 5 mm, as displayed in Figure 11a. This is a reliable result, as the Stanford dragon model is 200 mm by 100 mm by 150 mm in size. PLuM uses a reward-based metric, providing reliable results. Point cloud measurements that do not support the hypothesis are not rewarded using the lookup table. Error-based metrics have limited performance due to the clutter increasing the error between the point cloud and the model, as is evident in the performance of the ICP-based methods displayed in the maximum vertex error results. While Robust ICP performed well in the presence of no clutter, the registration procedure failed with increasing clutter. Increasing the point cloud measurements due to the addition of clutter slightly increased PLuM’s execution time due to the increase in the number of lookup operations for each hypothesis as observed in Figure 11b. The execution time of the PLuM algorithm is bi-proportional to the number of hypotheses and the size of the point cloud. The reward for each hypothesis needs to be calculated. Each hypothesis requires the reward lookup for every point cloud measurement. The data can be subsampled to increase the speed with larger point clouds, with minimal loss of accuracy and a significant decrease in execution time as desired. Section 6 demonstrates this process with uniform point cloud subsampling. All accuracy and execution time results are summarised in Table 4.

### 5.3. The Effect of the Lookup Table Uncertainty

The single configurable parameter in PLuM’s objective function is the measurement uncertainty, σ. This value aims to approximate and encompass all uncertainty in the process, including the geometry model mismatch, sensor registration uncertainty, and sensor measurement uncertainty. Figure 12a–c display examples of lookup tables for the Stanford Bunny generated at various uncertainty values. Solving the same problem as depicted in Figure 7a with varying lookup table uncertainties resulted in a maximum vertex error of less than 5 mm, as reported in Table 5 and Figure 12d. This demonstrated the robustness of the algorithm’s objective function to its only configurable parameter. The timeliness and memory constraints of the algorithm were unaffected, as the lookup table is generated offline and loaded before use.

### 5.4. The Effect of Lookup Table Resolution

The lookup table resolution is proportional to the size of the lookup table, i.e., the greater the resolution, the more reward values that need to be stored. Hardware constraints limit the size of a large lookup table, as it is loaded during execution. Figure 13a–c display lookup tables at 5 mm, 15 mm, and 20 mm resolutions, respectively. The sparsity and lack of high reward values fitting to the model are visible in the lookup tables for the 15 mm and 20 mm plots, compared to the shape of the Bunny outlined by the lookup table at a 5 mm resolution. Table 6 and Figure 13d display the maximum vertex error using the different lookup table resolutions for solving the registration problem from Figure 7a. There was minimum deviation in the results using a lookup table resolution from 1 to 10 mm, with a considerable 1764 times decrease in the size of the lookup table from 30 MB to 17 KB when the entries were scaled and saved as uint8_t variables. This demonstrated that millimetre accuracy can be achieved with sparser lookup table resolutions. The vertex error at 15 mm and 20 mm resolutions increased as there were fewer points with higher reward values, making the search for the correct pose difficult.

### 5.5. The Effect of Measurement Uncertainty

The scenarios above assumed zero sensor uncertainty in the point cloud measurements. Real-world sensors have measurement uncertainty. For example, the Velodyne ULTRA Puck documents a range accuracy of ±30 mm [12] and the Ouster OS-128 documents a precision of ±15–50 mm [13]. These are typical values that vary under range, temperature, and reflectivity conditions. Figure 14 illustrates the effect of increasing sensor uncertainty on the reliability and timeliness of the solution. The problem from Figure 7a is solved with increasing uncertainty in the point cloud measurements. The vanilla ICP algorithm was unable to register any of the scenarios accurately. The Robust ICP algorithm performed very accurately for most tests, but at the expense of significantly increased computation time. The PLuM algorithm performed with similar accuracy as displayed in Figure 14b, at execution rates more than 45 times faster than Robust ICP. Similar to the clutter scenario, the point cloud measurements are rewarded instead of using an error-based metric, allowing the best pose to be unaffected by the measurement uncertainty. Furthermore, the lookup table uncertainty can be selected to match the sensor uncertainty, but as shown in the previous section, this had a minimal effect on the result. Robust ICP has its merits at the expense of execution time. All accuracy and execution time results are summarised in Table 7.

## 6. Case Study: Haul Truck Pose Estimation

There has been considerable research in previous years to improve earth-moving equipment’s productivity, efficiency, and safety, as outlined by [46]. Significant efforts have been made to automate various aspects and components of mining practices to address these three challenges [47,48,49,50,51,52,53].

The productivity of open-pit mining is significantly impacted by the efficiency of the excavation load cycle. Inefficient truck spotting has been a historical problem at mines [54] and is currently an active research area, with automation offering substantial improvements in optimising the excavation load cycle. The Australian Coal Association Research Program (ACARP) has invested heavily in a vision to develop an autonomous mining shovel, with advancements published in the Shovel Load Assist Project report by Dudley et al. [55]. Many operator-assistance technologies have been deployed by companies such as Modular Mining [54] and Neptec [56] in an attempt to optimise the process. As the cycle is procedural and follows a fixed pattern, recent advancements focus on fully automating this process. A key ingredient in advancing towards fully autonomous loading requires the localisation of haul trucks during the excavation cycle relative to the rope shovel. This involves having the ability to estimate the pose of the haul trucks as they arrive for loading, are loaded, and then leave the local vicinity. Figure 15 displays typical scenes during the excavation cycle.

In an attempt to explore the possibility of automating the excavation load cycle and achieving real-time truck pose estimation, a rope shovel was mounted with two LiDARs to collect point cloud data and capture the movement of the trucks. GPS antennas with RTK corrections were also mounted on a haul truck to provide the ground truth for comparison purposes with any pose estimates. The goal was to accurately determine the pose of the truck at rates commensurate with the 20 Hz point cloud data in an attempt to have real-time truck pose estimation. This is required for real-time decision-making at the control layer. The following demonstrates the application of the PLuM algorithm to track the haul truck during a 90 s loading cycle. The truck tray was used as the known geometry as it is consistent between trucks and captured sufficiently well by the point cloud measurements. The entire truck was not used, as the suspension reacts when loaded with material and also deforms the wheels. The point cloud is sparse and unable to capture the fine detail on the side of the truck that would aid with registration.

Truck pose estimation is a challenging task due to the mining environment. Dusty environments contribute to spurious measurements and imperfect observation. Furthermore, the truck tray geometry becomes occluded as the tray fills with material during the loading procedure. This means the point cloud of the tray varies as the truck arrives for loading, is loaded, and then leaves the site. The authors previously demonstrated the inability of ICP to correctly register the truck tray when loaded with material in [11] with the same dataset, hence being unable to accurately determine the truck pose. The tray geometry is also subject to model mismatch due to general wear and degradation on the trucks under observation. The sensor has measurement uncertainty, and the 3D point cloud is dense and captures the constantly changing surrounding environment in addition to the truck. Typical scenes are illustrated in Figure 16, showing the high-density point cloud, the dig face, and sparse measurements on the truck body. Ideally, the pose estimation algorithm should not require any preprocessing segmentation of the point cloud.

Utilising the same CUDA-accelerated C++ scripts as in the previous section with a lookup table of the truck tray as displayed in Figure 17 and updated search algorithm parameters based on the environment, the haul truck pose estimates were reported in an accurate and timely manner.

The strength of using a reward-based metric allows for an accurate and reliable solution when the tray point cloud is occluded with material or in the presence of an expected model mismatch. Figure 18 compares a PLuM pose estimate to a ground truth obtained using RTK-GNSS at a particular scan during the excavation loading cycle. The pose estimate accurately matched the ground truth, with a maximum vertex error, $e_{max}$, of 0.158 m. The red line illustrates the maximum error in the fitting of a 14.5 m × 8 m ×4.5 m tray truck.

The point cloud data typically consists of more than 40,000 points arriving at 50 ms (20 Hz) intervals. The results in the previous section demonstrated an increase in computation time with high-density point clouds. As the goal was to provide truck pose estimates in real-time, the data was uniformly subsampled to reduce the point cloud’s size. Figure 19a displays the maximum vertex error results of a 90-second loading procedure where the point cloud was uniformly subsampled (5%). There was increased vertex error at the start and end of the cycle as the truck entered and left the field of view of the LiDAR. There were limited measurements on the truck at these times. The average vertex error was less than 200 mm as the truck was loaded. The results were computed in real-time at approximately 20 Hz, as displayed in Figure 19b. The decreased execution time at 30 s and 80 s occurred due to the reduced size of the recorded point cloud. This is approximately 126,000 times faster than the previously reported results by the authors using MSoE with no loss of accuracy.

ICP requires point cloud segmentation of the area of interest prior to the registration process. The registration algorithm is unable to handle raw point cloud data, and therefore, cannot satisfy the same testing conditions as for PLuM, where no preprocessing is required other than subsampling the point cloud for computational benefit only. Table 8 summarises the results with ICP and MSoE. The PLuM algorithm demonstrates reliable and timely results in a field robotics application using real-world data, overcoming the majority of challenges of model-based pose estimation detailed in Section 2.

The ground truth was from the GNSS-RTK solution, incorporating up to a 10 mm pose error using a NovAtel FlexPak 6 [57]. The reward values for the PLuM estimate and GNSS solution are plotted together in Figure 20a. The PLuM estimates have similar reward values and are slightly higher than the reward obtained by the GNSS solution. Figure 20b displays the cumulative distribution of the maximum vertex error. 78% of the pose estimates reported an error of fewer than 0.2 m and 90% less than 0.26 m. The data incorporated scans as the truck entered and left the site, corresponding to minimal measurements on the tray, leading to an increase in the estimation error. The ego-motion or the swing of the rope shovel contributed to the pose estimation error. At a maximum rope shovel swing speed of 15°/s, the sensor origin was displaced by up to 30 mm when completing a scan at 20 Hz. This shift in the sensor origin further displaced the point cloud measurements on the truck. Considering that the truck was an average of 10 m from the sensor and within a 105° field of view from the sensor’s origin, the first point cloud measurement on the truck was displaced 60 mm due to the sensor origin moving. These sources of error were not considered in the pose estimation calculation.

Figure 21 displays the effect of varying lookup table uncertainty values on the pose estimation maximum vertex error. Every 100th scan is plotted for visualisation purposes, with each colour representing the pose estimation accuracy using a different lookup table uncertainty. As shown, the lookup table uncertainty did not have a significant effect on the accuracy of the pose estimate solution. The PLuM objective function ensured that the maximum reward only corresponded to the correct pose of the geometry. The execution time was unaffected as the lookup table is loaded offline.

## 7. Conclusions

This paper proposed a pose estimation algorithm for known geometries in point cloud data that satisfies the criteria of being reliable and fast. Reliability was addressed by using a reward-based metric allowing for robustness against sensor uncertainty, clutter, and any mismatch between the known and the observed geometry. The results demonstrated superior accuracy over ICP-based methods, including the vanilla ICP algorithm, Fast and Robust ICP, and Sparse ICP, for 6-DOF registration problems in the presence of significant measurement uncertainty and clutter. The execution time was reduced by using pre-calculated lookup tables to replace complex geometric operations such as measurement-point-to-model raycasting. The results demonstrated a timely solution in all test cases utilising CUDA for parallelising the hypothesis evaluation. The execution time was bi-proportional to the size of the point cloud and the hypothesis set.

PLuM is an objective function requiring a search algorithm to provide pose estimates. The search algorithm is independent of the objective function and open to exploration. Use of the particle filter in the results demonstrated adequacy in locating global extrema. The single configurable parameter of the PLuM algorithm is the lookup table uncertainty, σ, which was shown to be robust to a range of values. The input point cloud requires no segmentation. However, it may need to be subsampled to observe faster execution times.

The significant contribution of this paper was providing a solution to the model-based pose estimation problem that is both robust and fast. Majority of the existing solutions provide a compromise between accuracy and speed. The motivation for developing the PLuM algorithm was to provide a simple implementation that is use-case agnostic. The only difference in the configuration between the Stanford model tests and the truck pose estimation was the model lookup table and the search algorithm parameters.

Benchmark tests using the Stanford Bunny and Dragon models resulted in accurate and timely results in the presence of sensor uncertainty, clutter, and poor initial alignment. Furthermore, the PLuM algorithm was demonstrated in a field robotics application to accurately track haul trucks in real-time throughout the excavation loading procedure. This is a challenging task due to the unpredictable and unordered mining environment and the significant size of the point cloud, which is open to interpretation. Previous attempts using ICP provided inaccurate results, as the truck tray geometry became occluded with material during loading [11]. However, using a reward-based metric overcame this challenge.

PLuM provides a reliable and timely solution to answer the question of *where is it*? However, the algorithm always provides a solution, even when there is no model in the scene. The question of *is a model present in the scene?*, or *is there more than one model present in the scene?* remain unanswered. Future research will examine providing a confidence measure to determine if a model, or more than one model, is present in the environment. The results can also be extended to estimating the pose of multiple articulated bodies, such as a rope shovel’s dipper and the dipper’s door geometry. It is expected that this will require two lookup tables, with the computation time expected to double.

## Figures and Tables

**Figure 1 sensors-23-03085-f001:**
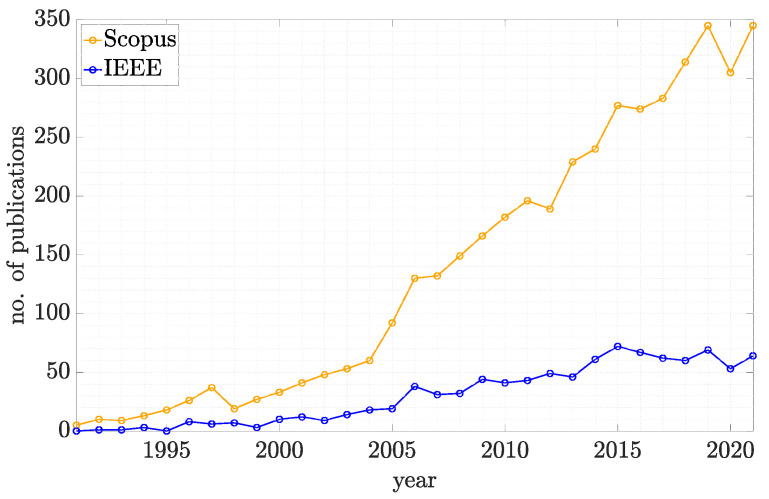
The pose estimation research space is saturated with variants or the usage of the Iterative Closest Point algorithm and continues to grow exponentially. The data was recorded by searching for the keywords *iterative closest point* in the title and/or abstract. The plot is adapted from [25].

**Figure 2 sensors-23-03085-f002:**
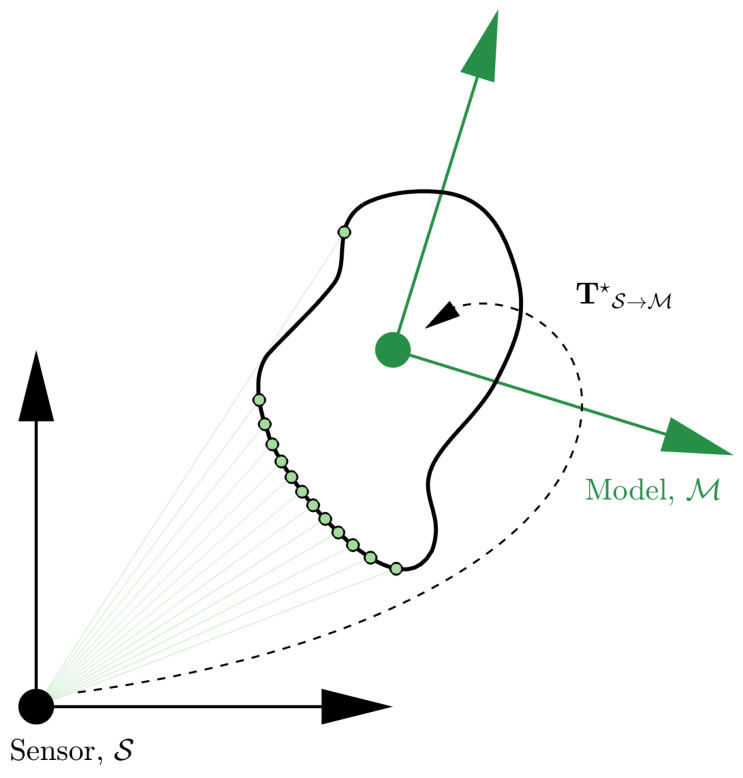
The model-based pose estimation problem is illustrated using a known geometry displayed in black and a point cloud in green. The goal is to determine the homogeneous transformation $T_{S\to M}^{🟊}$ that is most likely to reproduce the observed point cloud measurements.

**Figure 3 sensors-23-03085-f003:**
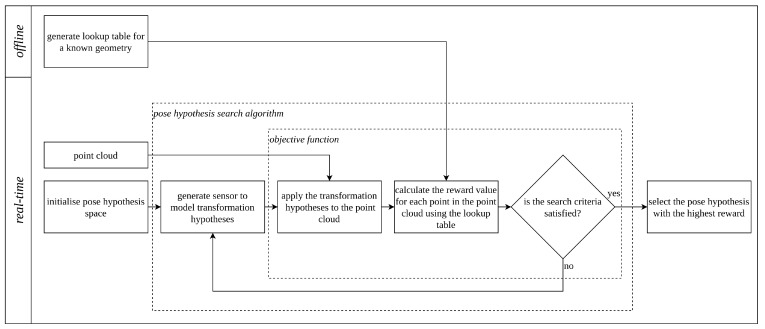
A flowchart summarising the entire process. The search algorithm uses the objective function to determine the best pose estimate. The contribution of this paper is the PLuM objective function. This can be used with a search algorithm of choice. Search algorithms have a termination condition, commonly a change in gradient for gradient-based solvers or the number of iterations for a particle search.

**Figure 4 sensors-23-03085-f004:**
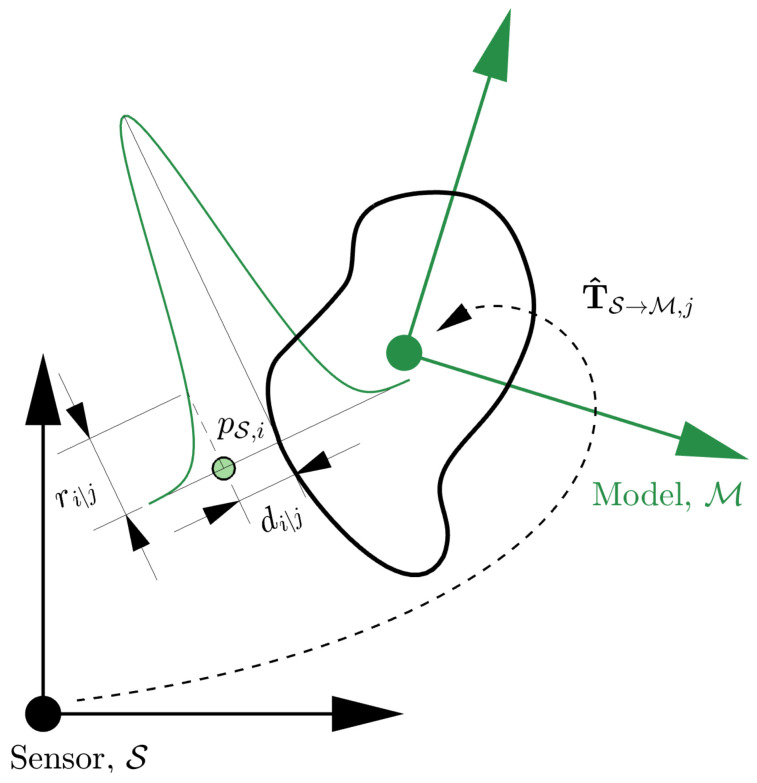
Consider a geometry model that has been located relative to the sensor with the *j*-th pose hypothesis, ${\hat{T}}_{S\to M,j}$. The *i*-th point in the sensor frame, $p_{S,i}$, has a distance of $d_{i|j}$ to the model’s surface. This would result in a reward of $r_{i|j}$ when evaluated with the Gaussian reward function.

**Figure 5 sensors-23-03085-f005:**
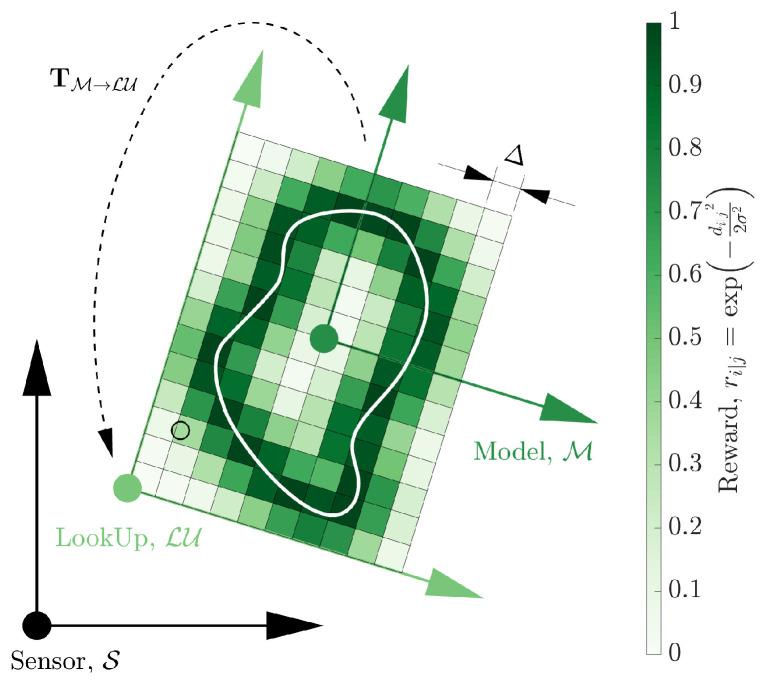
A lookup frame, LU, is selected such that a bounding region of pixels (or voxels in 3D) around the model can be described by precalculated distances to the model surface. The pixels are discretised at a resolution of Δ, as shown. The lookup frame is rigidly attached to the model coordinate frame (with transform $T_{M\to \mathrm{LU}}$) such that it moves with the hypotheses of the model pose.

**Figure 6 sensors-23-03085-f006:**
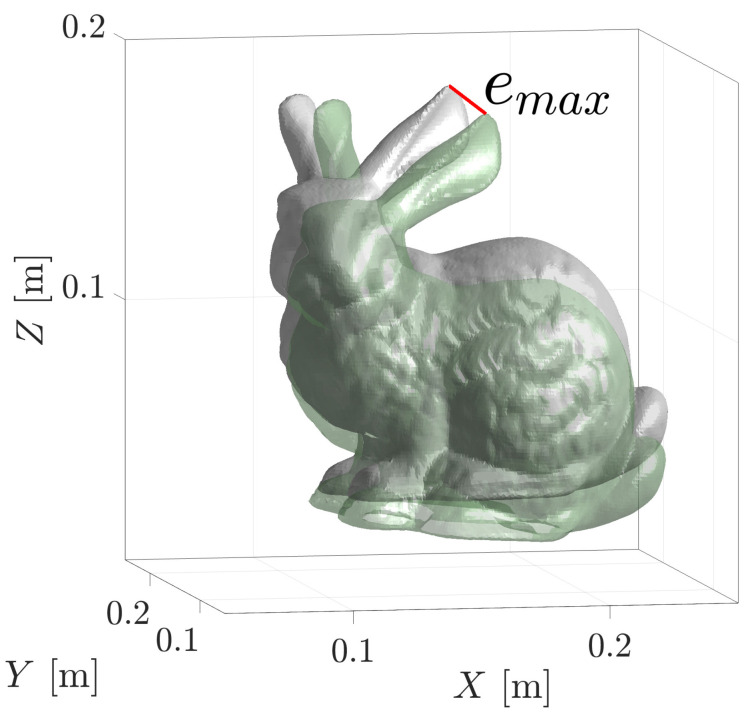
The accuracy of a pose estimate is measured using the maximum vertex error metric, $e_{max}$, as displayed by the red line. This distance is the maximum displacement of the geometries between the estimate (green) and the correct pose (white).

**Figure 7 sensors-23-03085-f007:**
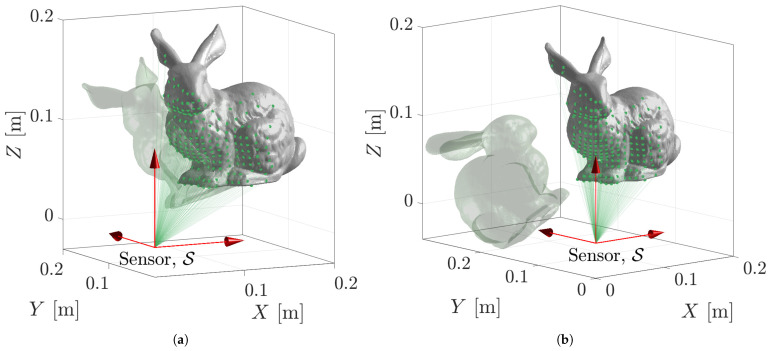
Two candidate problems with the same ground truth and point cloud consisting of 202 measurements. The different seeds, or initial guesses, are displayed in green. (**a**) Provides significant overlap between the point cloud and the initial position, whereas (**b**) is a harder problem having an initial position with no overlap.

**Figure 8 sensors-23-03085-f008:**
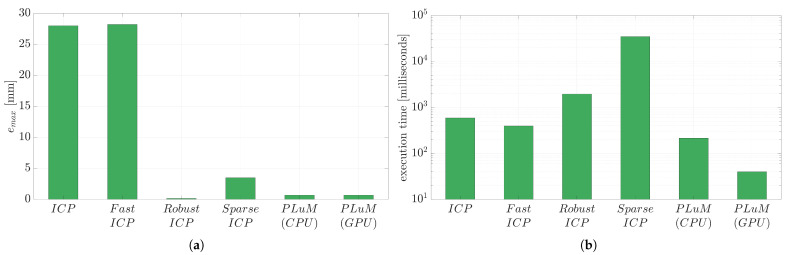
The reported results for the problem depicted in Figure 7a. (**a**) Displays the maximum vertex error ($e_{max}$), and (**b**) compares the execution time.

**Figure 9 sensors-23-03085-f009:**
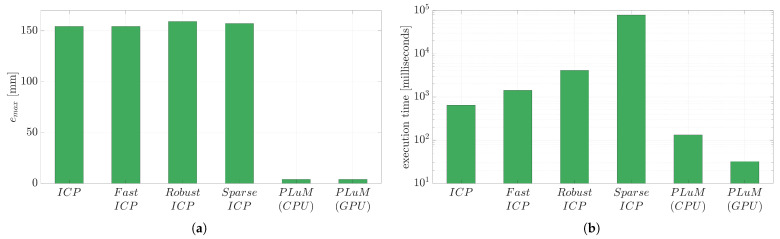
The reported results for the problem depicted in Figure 7b. (**a**) Displays the maximum vertex error ($e_{max}$), and (**b**) compares the execution time.

**Figure 10 sensors-23-03085-f010:**
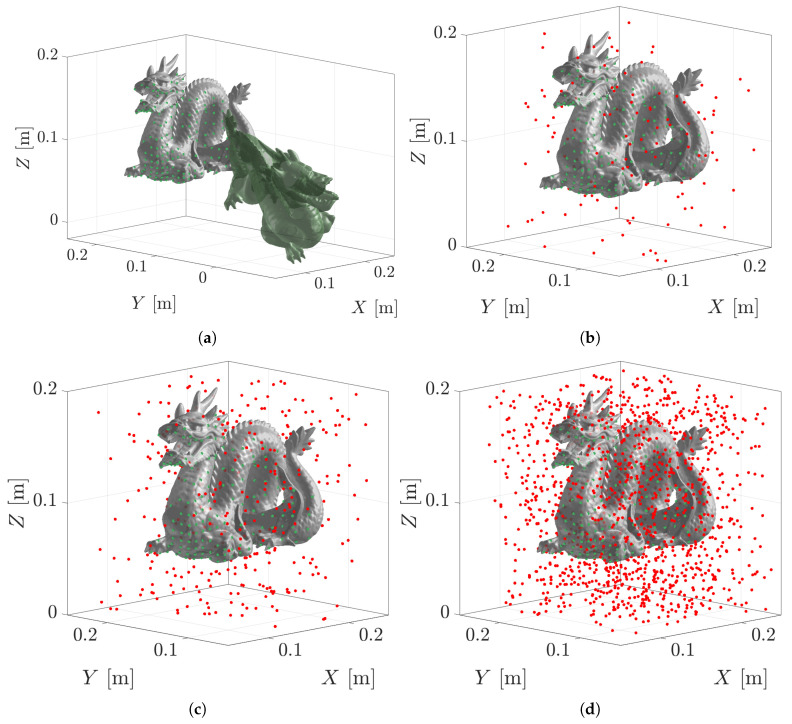
Real-world scenarios have unwanted measurements in the scene. The original point cloud is green, and the clutter measurements are red. The true pose of the dragon is $T_{S\to M}^{🟊}=\left[1.57,0,0,0.18,0.16,0\right]$. (**a**) Displays a clutter-free point cloud with 187 measurements. The initial pose of the dragon is ${\hat{T}}_{S\to M}=\left[1.2,-0.5,1.5,0.25,0.1,-0.03\right]$ and is displayed in dark green. (**b**) Has 384 measurements, of which 50% are random clutter. (**c**) Has 624 measurements, of which 70% are random clutter. (**d**) Has 1870 measurements, of which 90% are random clutter.

**Figure 11 sensors-23-03085-f011:**
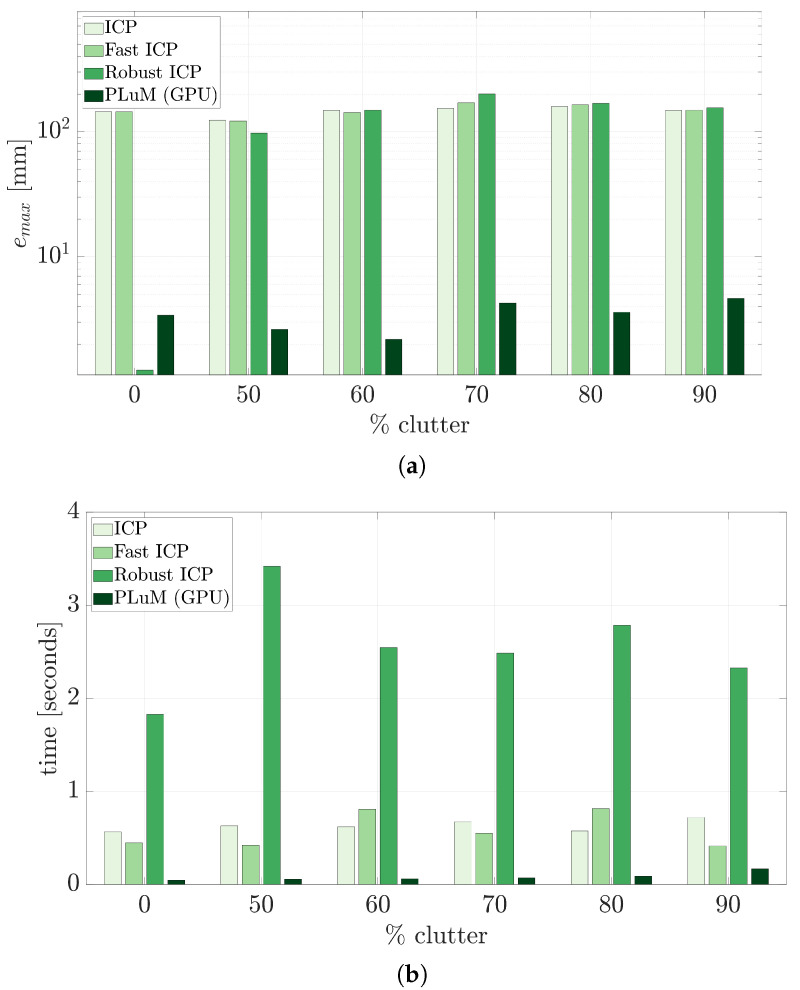
(**a**) Displays the performance of the registration algorithms with increasing random clutter. PLuM provides reliable results with varying amounts of clutter in the point cloud, as measurements that do not fit the model do not penalise the correct hypothesis. (**b**) Shows the slightly increasing execution time of the PLuM algorithm as the point cloud size increases due to the added clutter. The timeliness of the other methods can be disregarded, as they fail to provide an accurate registration result.

**Figure 12 sensors-23-03085-f012:**
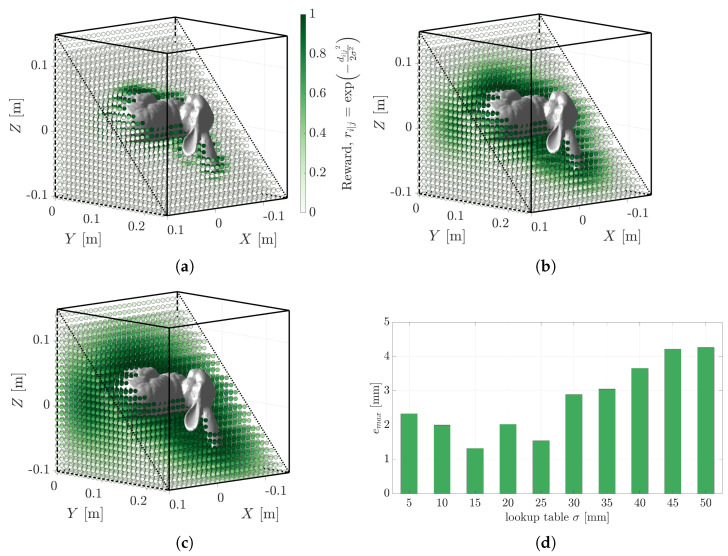
Visual representation of the lookup table for the Stanford Bunny at various measurement uncertainties: (**a**) 10 mm, (**b**) 30 mm, and (**c**) 50 mm. (**d**) Displays the minimal effect on the maximum vertex error reported by the PLuM algorithm in solving the registration problem depicted in Figure 7a using increasing lookup table uncertainties.

**Figure 13 sensors-23-03085-f013:**
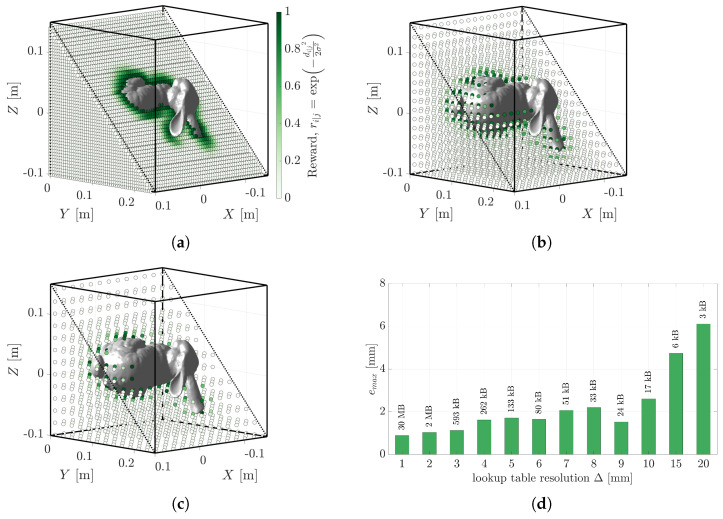
Visual representation of the lookup table for the Stanford Bunny at various lookup table resolutions: (**a**) 5 mm, (**b**) 15 mm, and (**c**) 20 mm. (**d**) Displays the minimal effect on the maximum vertex error reported by the PLuM algorithm in solving the registration problem depicted in Figure 7a using decreasing lookup table resolution. The size of each lookup table is displayed on top of each bar when the entries were scaled and saved as uint8_t variables.

**Figure 14 sensors-23-03085-f014:**
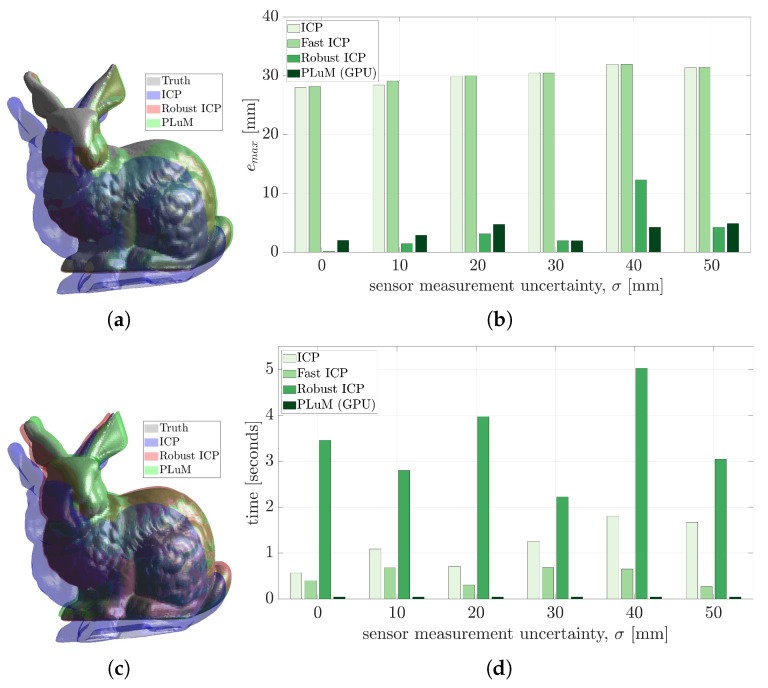
(**a**,**c**) Show example registration results for the problem in Figure 7a with 10 mm and 30 mm measurement uncertainty, respectively. The effect of adding sensor measurement uncertainty on the maximum vertex error is displayed in (**b**), and the corresponding execution times in (**d**).

**Figure 15 sensors-23-03085-f015:**
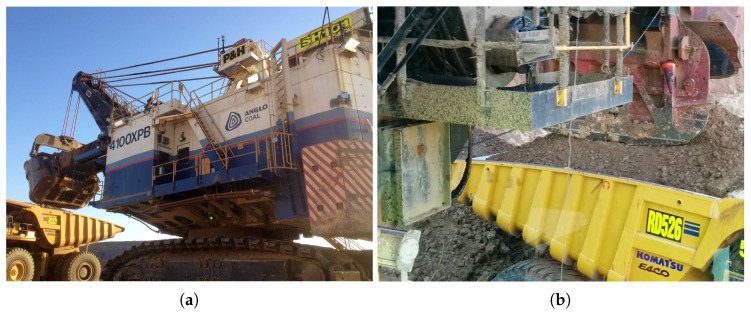
(**a**) Shows a rope shovel loading a haul truck with material during the excavation cycle. (**b**) Displays the truck’s tray being occluded with material.This creates a challenging environment when using the truck’s tray as the known geometry for point cloud registration.

**Figure 16 sensors-23-03085-f016:**
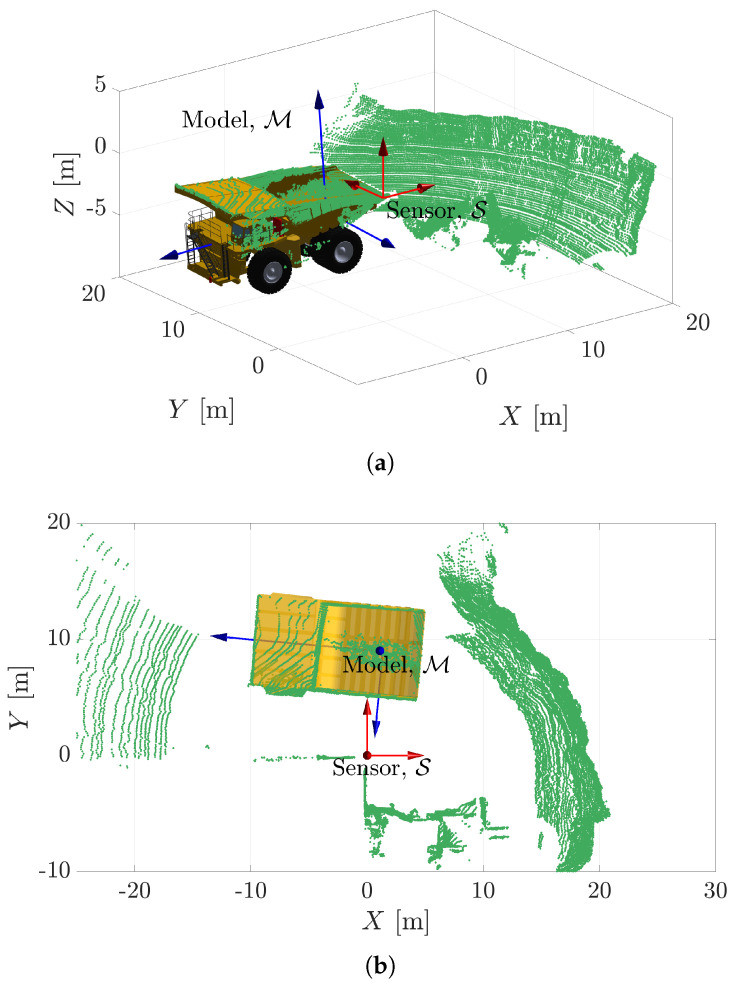
(**a**,**b**) Show the typical scene at a rope shovel operation captured by the sensor. Some measurements on the truck are occluded by the material during the load cycle. The goal is to accurately determine the pose of the truck (M) at rates commensurate with the 20 Hz LiDAR (S) mounted at a registered position on the rope shovel.

**Figure 17 sensors-23-03085-f017:**
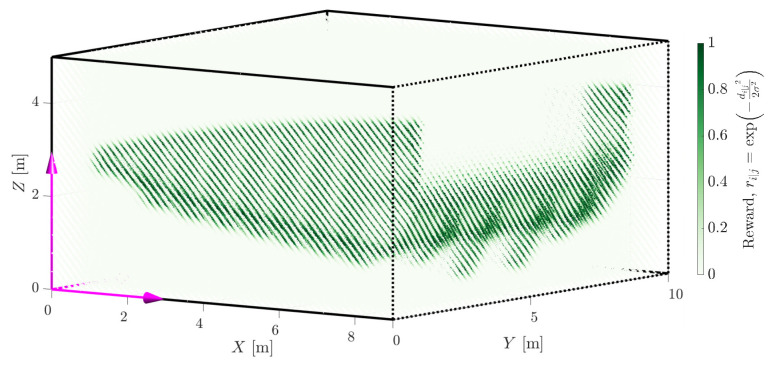
A lookup table generated for the truck tray with an uncertainty of 100 mm. The lookup table has been truncated, and every 1000th point is shown for visualisation purposes. The lookup frame (LU) is displayed at the origin.

**Figure 18 sensors-23-03085-f018:**
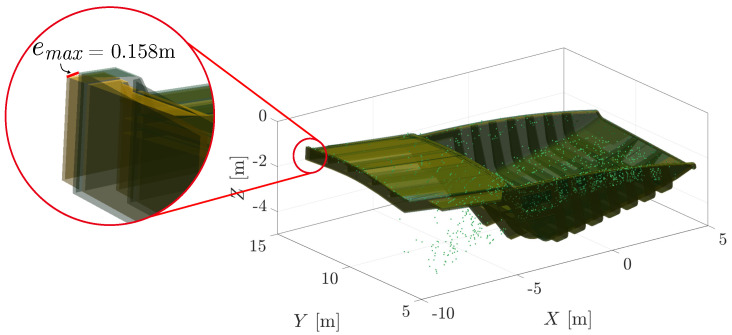
A sample result of a PLuM truck pose estimate fitting to the GPS estimate. The maximum vertex error in the top right corner of the tray is 0.158 m. The error is minimal in comparison to the large size of the geometry.

**Figure 19 sensors-23-03085-f019:**
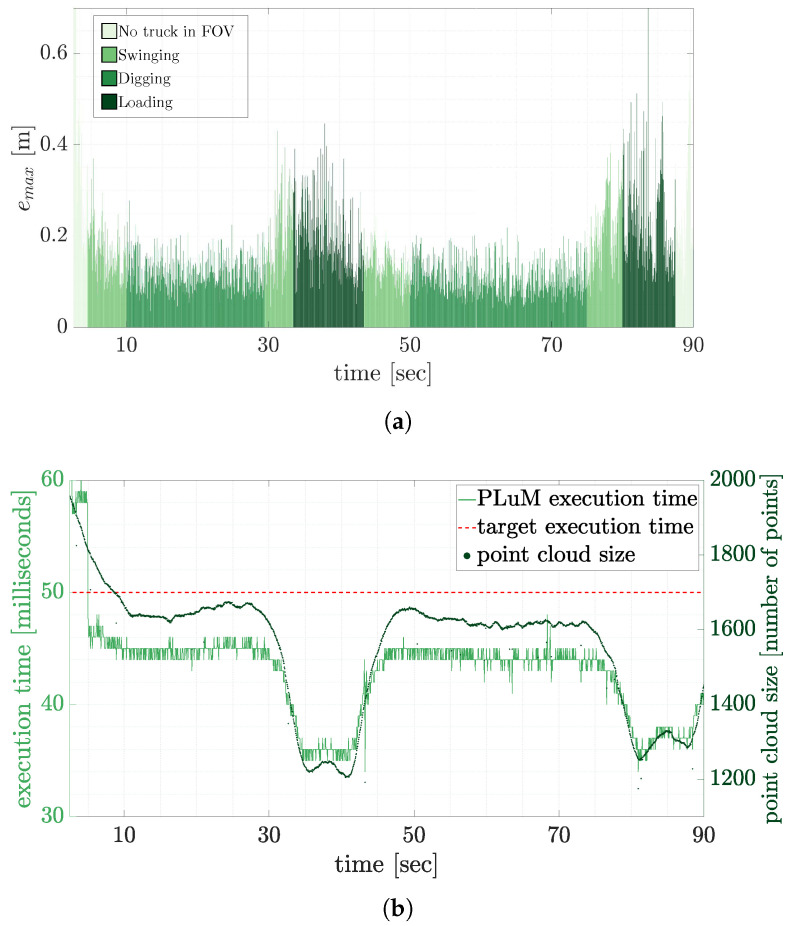
The PLuM truck pose estimation results using point cloud data subsampled using a factor of 20, i.e., the original point cloud reduces from 40,000 to 2000 measurements. The maximum vertex error for a 90 s loading cycle is reported in (**a**), and the corresponding execution time for each pose estimate is displayed in (**b**).

**Figure 20 sensors-23-03085-f020:**
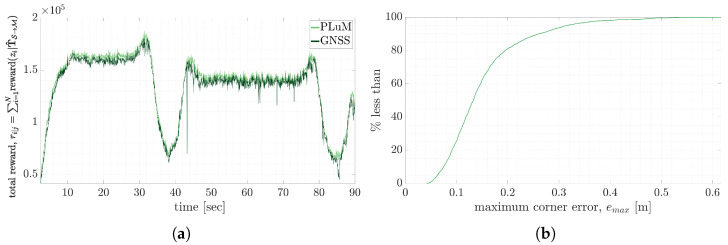
(**a**) Compares the maximum reward for the PLuM pose estimates and the reported GNSS solution. The reward is similar, with the PLuM algorithm reporting a slightly higher reward on average. (**b**) Displays a cumulative distribution function for the maximum vertex error for the 90 s loading cycle.

**Figure 21 sensors-23-03085-f021:**
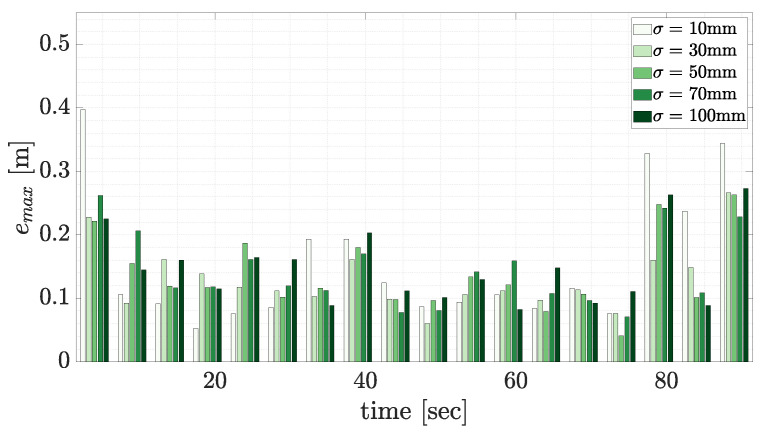
The effect of the lookup table uncertainty on the maximum vertex error using PLuM. The result for every 100th scan is displayed for visualisation purposes. Each darker colour in a group represents an increasing lookup uncertainty, σ, from 10 mm to 100 mm. The lookup tables were generated with a resolution of Δ=10 mm.

**Table 1 sensors-23-03085-t001:** Summary of point cloud-to-model registration algorithms.

Method	Strengths	Limitations
Iterative Closest Point (ICP) [10]	Accurate 6-DOF registration with noise-free point clouds. The algorithm is independent of shape representation.	The results strongly depend on initial alignment. The registration process requires preprocessing such as point cloud segmentation and does not provide real-time results. The solution tends to converge to local extrema and is not robust with clutter and noisy data.
Sparse ICP [27]	Improves the performance of ICP with noisy and incomplete data.	The algorithm is computationally expensive. The solution converges to local extrema when there is minimal overlap between the two datasets.
Go-ICP [28]	The first globally optimal ICP algorithm, providing accurate results when good initialisation is not possible.	The algorithm is limited to scenarios where real-time performance is not critical.
Fast and Robust ICP [29]	The algorithm provides accurate results on noisy datasets and with some clutter. The performance offers a significant improvement over the generic ICP algorithm.	The performance requires good initialisation. The Fast ICP variation compromises accuracy for speed. The Robust ICP variation compromises speed for accuracy.
Maximum likelihood estimation variations of ICP [30,31,32,33]	This class of algorithms performs well without good initialisation. They provide accurate registration in the presence of noise, outliers, and missing measurements.	The algorithms tend to converge to local solutions, and many do not provide real-time results.
Maximum Sum of Evidence (MSoE) [2]	This reward-based method provides an accurate point cloud-to-model registration and performs well without good initialisation. The results are accurate in noisy, occluded, and cluttered environments.	The algorithm is computationally expensive and does not provide real-time results.
Geometric deep-learning-based methods [34,35,36]	These methods provide accurate performance in scenarios similar to the trained data.	The performance is not deterministic with unseen cases, and it is difficult to trace errors. The methods require sufficient training data.

**Table 2 sensors-23-03085-t002:** The maximum vertex error and execution time for each method in the problem depicted in Figure 7a. The best results are displayed in bold.

	ICP	Fast ICP	Robust ICP	Sparse ICP	PLuM (CPU)	PLuM (GPU)
$e_{max}$ (mm)	27.98	28.18	**0.14**	3.47	2.89	2.89
*t* (ms)	566	455	1813	34174	118	**25**

**Table 3 sensors-23-03085-t003:** The maximum vertex error and execution time for each method in the problem depicted in Figure 7b. The best results are displayed in bold.

	ICP	Fast ICP	Robust ICP	Sparse ICP	PLuM (CPU)	PLuM (GPU)
$e_{max}$ (mm)	154.36	154.37	159.21	157.27	**3.89**	**3.89**
*t* (ms)	646	1425	4112	78566	132	**32**

**Table 4 sensors-23-03085-t004:** The maximum vertex error and execution time for each method. The best results are displayed in bold.

% Clutter $e_{\max}$ (mm) (*t* (s))	ICP	Fast ICP	Robust ICP	PLuM (GPU)
0	144.47 (0.56)	145.54 (0.45)	**1.24** (1.83)	3.42 (**0.05**)
50	123.43 (0.63)	121.93 (0.42)	97.65 (3.42)	**2.63** (**0.06**)
60	147.63 (0.62)	142.16 (0.81)	142.16 (2.55)	**2.19** (**0.06**)
70	153.46 (0.67)	169.81 (0.55)	200.32 (2.49)	**4.27** (**0.07**)
80	158.64 (0.57)	163.93 (0.81)	168.60 (2.79)	**3.61** (**0.09**)
90	147.52 (0.72)	148.45 (0.41)	155.08 (2.33)	**4.66** (**0.17**)

**Table 5 sensors-23-03085-t005:** The maximum vertex error for varying values of the lookup table uncertainty.

σ (mm)	5	10	15	20	25	30	35	40	45	50
$e_{max}$ (mm)	2.33	2.00	1.32	2.03	1.54	2.89	3.05	3.66	4.22	4.27

**Table 6 sensors-23-03085-t006:** The maximum vertex error for varying values of the lookup table resolution.

Resolution (mm)	1	2	3	4	5	6	7	8	9	10	15	20
$e_{max}$ (mm)	0.89	1.03	1.13	1.61	1.71	1.65	2.07	2.20	1.53	2.61	4.73	6.11

**Table 7 sensors-23-03085-t007:** The maximum vertex error and execution time for each method. The best results are displayed in bold.

Uncertainty (mm) $e_{\max}$ (mm) (*t* (s))	ICP	Fast ICP	Robust ICP	PLuM (GPU)
0	27.98 (0.57)	28.18 (0.40)	**0.14** (3.46)	2.00 (**0.05**)
10	28.45 (1.09)	29.10 (0.68)	**1.44** (2.80)	2.89 (**0.05**)
20	29.92 (0.71)	30.00 (0.30)	**3.11** (3.97)	4.73 (**0.05**)
30	30.49 (1.26)	30.48 (0.68)	**1.97** (2.23)	1.98 (**0.05**)
40	31.96 (1.80)	31.96 (0.65)	12.29 (5.03)	**4.24** (**0.05**)
50	31.38 (1.68)	31.39 (0.27)	**4.21** (3.04)	4.86 (**0.05**)

**Table 8 sensors-23-03085-t008:** The maximum vertex error and execution time in comparison to other methods.

		$e_{\max}$ (m)				
	Method	0.1	0.2	0.4	0.6	t/pose (s)
**% estimates less than**	**ICP**	unable to solve under the test conditions				
	**MSoE**	3	55	95	99	6055
	**PLuM**	25	78	96	99	0.048

## Data Availability

All tests in Section 5 use the publicly available triangulated geometries from The Stanford 3D Scanning Repository.

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
