# Peer review of "Real-Time 6-DOF Pose Estimation of Known Geometries in Point Cloud Data"

_sensors, 2023, doi:10.3390/s23063085_

Round 1

Reviewer 1 Report

The idea of the study "Real-Time 6-DOF Pose Estimation of Known Geometries in Point Cloud Data" is interesting. However, the following remarks should be incorporated:

1. The introduction part is well written. However, it can be polished in multiple ways (include a process flow that indicates the overall summary [graphical abstract]).

2. The literature review part should be summarized in a tabular form containing the major studies along with their strength and limitations.

3. Major contributions should be added that are to be addressed with the proposed approach.

4. Equation (2) does not contribute much as this is the universal one.

5. Results parts are promising.

6. Authors have contributed enough time to discuss the results. However, they are advised to mention their performance quantitatively (in tabular form) and compare it with state-of-the-art methods.

Reviewer 2 Report

This paper proposes a pose-tracking algorithm from point cloud measurements for robot perception applications. The algorithm known as Iterative Closest Point is a probabilistic reward-based objective function robust to measurement uncertainty and clutter. By using lookup tables, the algorithm also becomes computationally efficient. The algorithm performs with millimeter accuracy in benchmark tests. The algorithm also works well in real-time haul truck pose estimation using point clouds from a LiDAR. 

This reviewer believes this paper is well structured, and the results support the author's claims; therefore, I suggest accepting this paper. 

If possible, I suggest that the authors make the code available.

Reviewer 3 Report

This paper proposes a pose estimation algorithm for known geometries in point cloud data. According to the expression, organization, and experiment, I think there still have many problems in the paper.

1. In the Introduction, the motivation and contribution of the method should be highlighted.

2. Group related works in families. The advantages and limitations of these methods are not well summarized, and should be expressed with more emphasis on the difference with the proposed method.

3. This paper has two contributions, i.e. reward-based metrics and lookup tables. However, the used reward-based metrics looks the same as the proposed method in the literature [2]. Explain the difference between the two methods.

4. The flowchart of the proposed method should be presented.

5. The initial pose in Influence of the initial pose estimate (5.1) is specified. Whether it can be set randomly within a certain range.

6.Whether the execution time contains lookup table generation time? If not, whether the all execution time of the proposed method still has an advantage?

7. The comparison should include more state-of-the-art methods to demonstrate the effectiveness of the proposed method.

8. To verify the performance of these selected models from different aspects, more datasets with different characteristics should be added.

Round 2

Reviewer 3 Report

In the new version of this paper, the authors provide satisfactory replies corresponding to my questions including revision and additional analysis of experiments which show more detailed performance of the proposed method.
Thus I think this paper can be accepted after some improving of the readability.